# Beyond Amyloid Fibers: Accumulation, Biological Relevance, and Regulation of Higher-Order Prion Architectures

**DOI:** 10.3390/v14081635

**Published:** 2022-07-27

**Authors:** Wesley R. Naeimi, Tricia R. Serio

**Affiliations:** Department of Biochemistry and Molecular Biology, The University of Massachusetts Amherst, Amherst, MA 01003, USA; wnaeimi@umass.edu

**Keywords:** amyloid, amyloidosis, prion, [*PSI*^+^], fibril, protofibrils

## Abstract

The formation of amyloid fibers is associated with a diverse range of disease and phenotypic states. These amyloid fibers often assemble into multi-protofibril, high-order architectures in vivo and in vitro. Prion propagation in yeast, an amyloid-based process, represents an attractive model to explore the link between these aggregation states and the biological consequences of amyloid dynamics. Here, we integrate the current state of knowledge, highlight opportunities for further insight, and draw parallels to more complex systems in vitro. Evidence suggests that high-order fibril architectures are present ex vivo from disease relevant environments and under permissive conditions in vivo in yeast, including but not limited to those leading to prion formation or instability. The biological significance of these latter amyloid architectures or how they may be regulated is, however, complicated by inconsistent experimental conditions and analytical methods, although the Hsp70 chaperone Ssa1/2 is likely involved. Transition between assembly states could form a mechanistic basis to explain some confounding observations surrounding prion regulation but is limited by a lack of unified methodology to biophysically compare these assembly states. Future exciting experimental entryways may offer opportunities for further insight.

## 1. Introduction

Normal cellular homeostasis requires maintenance of a correctly folded proteome (i.e., proteostasis), which is enabled by a network of tightly regulated protein-quality control factors including chaperones and proteases [1]. When the efficiency of this network is reduced, normally stable proteins adopt off-pathway conformational folds that promote their aggregation. Changes in fold and assembly state are known to alter protein recognition by the proteostasis network [2], and amplification of the network in response to emergence of these species can generally promote restoration of proteostasis [3]. This adaptive response, however, has its limits, particularly in the presence of recalcitrant substrates whose accumulation can dramatically alter cellular phenotypes, promote toxicity and disease states, and ultimately lead to death [4]. Intriguing observations suggest that structural changes beyond a two-state, soluble-to-aggregate transition contribute to the biological consequences associated with misfolding of these substrates (Figure 1) [5,6,7,8,9,10,11,12,13,14], but a clear understanding of the interconnections between protein states in vivo, their regulation by the proteostasis network, and the resulting impact of this interplay on phenotype has yet to emerge. 

Amyloidogenic proteins provide an excellent model to elucidate this interplay. These proteins can access alternative conformations under physiological conditions and assemble into aggregates that are associated with either new functional or disease states. Each of these phenotypic states can often exist within a range of distinct severities and stabilities dictated by the underlying fold of the protein determining them [15]. The persistence of these aggregates results from a combination of their high thermodynamic, kinetic, and metabolic stability, as well as a unique property: the capability to template the conversion of natively folded versions of that protein into the same amyloid conformation [16]. These characteristics are conferred by a common underlying cross-β structure, which includes tight interfaces devoid of water and a repeating β-sheet that presents a templating surface [17]. Remarkably, a single protein can assemble into distinct amyloid forms, known as polymorphs. These structures are distinguished by the packing of β-sheets and/or the segment(s) of the protein forming the β-sheets within the amyloid core. In either case, different side chains are accessible on the surface, and this characteristic, along with the high-energy barriers between the polymorphs, creates a spectrum of self-templating structures that determine biological strains or variants in vivo [18]. The conformational conversion and assembly of proteins into the amyloid state has been associated with more than 50 human diseases, including Alzheimer’s, Huntington’s, and Parkinson’s diseases, Type II diabetes, and familial hypertension [19]. Despite the persistence of this state in vivo, components of the proteostasis network have been identified as modulators of amyloid assembly in vitro and disease appearance and progression in vivo, revealing some level of regulation [20].

Prion proteins are a particularly intriguing sub-class of amyloidogenic proteins: the properties of the amyloid state of prion proteins permit its transmission from cell to cell and organism to organism [21]. In the case of the mammalian prion protein PrP, appearance of the amyloid form denoted PrP^Sc^ leads to progressive and infectious neurodegenerative disease for which there is currently no treatment or cure [22]. In contrast, the amyloid state of other prion proteins such as the nine identified in the yeast *Saccharomyces cerevisiae* are tolerated well by cells and give rise to heritable phenotypes associated with either a loss or gain of function for the respective aggregated protein, influencing multifaceted aspects of cellular physiology from transcriptional regulation to nutrient utilization [23]. While both PrP and yeast prions are acted upon by components of the proteostasis network in vitro and in vivo [24], modulation of the activity of molecular chaperones can actually reverse amyloidogenesis for yeast prions, returning protein previously found in amyloid aggregates to the soluble state [25]. Thus, these stably propagated amyloid states in yeast offer a particularly tractable experimental model in which the connections between amyloid aggregation state, phenotype, and their regulation can be elucidated. 

The first identified and arguably most widely studied yeast prion is [*PSI*^+^] [26]. The [*PSI*^+^] prion is determined by the amyloid state of the translation termination factor Sup35, which results in a partial loss of function and the associated read-through of stop codons [27,28], with the prion-free and fully functional form of Sup35 associated with the [*psi*^−^] state [29]. The emergence of a stable [*PSI*^+^] state in growing yeast cultures exists within a balanced cycle of Sup35 expression, conversion into amyloid aggregates, fragmentation of these aggregates by chaperones, and transmission of these aggregates to daughter cells at cell division to maintain both the amyloid form and sufficient Sup35 activity to support viability (see Figure 2).

Like other amyloidogenic proteins, Sup35 can access amyloid polymorphs to create a “cloud” of variants in vivo from which individual conformations can be amplified to form stably propagating amyloid states under different proteostatic conditions, presumably aligned with their optimal replication rates [30]. The selected prion variants are distinguished by their defining amyloid cores [31,32], which present unique templating surfaces that determine the rate at which they convert native-state protein. In addition, these unique structures determine the kinetic stability of the aggregates and thereby the rates at which they are fragmented by molecular chaperones and the number of templating surfaces and transmissible aggregates [33]. While it is clear that these properties alter the severity and stability of the prion-associated phenotypes, the current molecular model contextualizes these phenotypic differences through the accumulation and size of an amyloid aggregate that is presumed to be monotypic [16]. Nonetheless, a variety of ultrastructural forms of Sup35 have been detected both in vivo and in vitro [14,34], and if transitions between these aggregate architectures are occurring, these species should contribute to further defining the key parameters for prion biology (see Figure 1). The relationship between these forms, their modulation by the proteostasis network, and their specific contributions to the appearance, persistence, and curing of the prion state are emerging but far from proven. Here, we integrate the current state of knowledge, highlight opportunities for further insight, and draw parallels to more complex systems.

## 2. The Functional Properties of Amyloids Assembled In Vitro or In Vivo Are Similar but Distinct

Much insight into the biophysical architecture of amyloid has been provided by in vitro studies where amyloid structures are assembled under permissive conditions from recombinant purified protein, typically a fragment of full-length protein that is necessary and sufficient to undergo this transition. The resulting polymers are linear aggregates known as protofibrils. Such studies have uncovered surprising similarities across a range of different amyloid-forming proteins where amyloid protofibrils are composed of monomers assembled linearly by stacking similarly folded domains into a highly protected amyloid core [35,36]. The dimensions of these protofibrils are largely consistent across multiple amyloid-forming proteins and sufficient to impart the templating function characteristic of the amyloid structure [37]. The remaining domains of the protein then sit tethered at the surface of the protofibril [38], at times globular in nature and functionally folded [39,40]. How relevant are in vitro-assembled fibrils to the structures which propagate in vivo? Studies examining both the biological and biochemical properties of these aggregates have provided insight into this crucial question and reveal that caution in both experimental design and interpretation is warranted.

For mammalian prions and prion-like proteins, in vitro-assembled amyloid, while sharing some biochemical similarities to ex vivo amyloid of the same protein, often fails to recapitulate the biological activity of the latter. For example, proteinase K-resistant PrP fibril preparations produced entirely from recombinant PrP (rec-PrP) alone can lack any detectable infectious activity in either cell culture or rodent bioassays [41]. However, proteinase K-resistant PrP produced from rec-PrP is infectious when seeded by diseased ex vivo brain homogenate using protein misfolding cyclic amplification (PMCA), a method that uses rounds of ultrasound treatment to fragment PrP polymers, thereby increasing the number of templates for PrP conversion to PrP^Sc^ [42]. Nonetheless, the observed infectivity of the PMCA-derived material is still much lower than that of ex vivo material itself. Further work surrounding the creation of these “synthetic prions” has identified polyanions and RNA as key cofactors required for forming infectious PrP^Sc^ in vitro [43,44], though co-factors alone appear insufficient in producing reproducibly infectious material without PMCA [45]. Subjecting rec-PrP to PMCA in the presence of the synthetic polyanion palmitoyloleoylphosphatidylglycerol (POPG) and liver-derived RNA is capable of producing PrP^Sc^ with infectivity levels similar to that of natural prions [46] through both intraperitoneal [46] and oral [47] routes of infection. How does PMCA and the presence of cofactors alter rec-PrP to modulate its infectivity? 

One possible explanation is structural differences between fibrils. Indeed, structural analysis of rec-PrP subjected to PMCA with or without the addition of co-factors found infectious preparations to contain smaller oligomeric species, while non-infectious material contained longer fibers [48], although the relative abundance of these two species was not assessed. At the ultra-structural level, non-infectious PrP fibrils produced from rec-PrP were not found to form the paired protofilament structures identified by cryo-EM in ex vivo preparations from diseased brain [41,49], suggesting that high-order architecture may contribute directly to biological activity in vivo. Similar observations have been made across many amyloid disease states, with ex vivo material displaying a higher-order architecture than in vitro-assembled fibrils for α-synuclein (α-syn) [50] and amyloid A [51]. In contrast, more diversity in morphology was found for amyloid-light chain (AL) fibrils produced in vitro than those isolated from AL amyloidosis-diseased human heart tissue [52]. Structural differences in the helical twist direction and sensitivity to proteinase K have also been observed between Aβ fibrils produced in vitro and those isolated ex vivo from human brain [53], while heparin-induced recombinant tau fibrils produced in vitro have smaller amyloid cores composed of different repeat compositions compared to those attained from Alzheimer’s- and Pick’s-diseased tissues [54]. While these correlations have been observed, the causal relationship of structural differences to biological impact, beyond the selective pressures that supported the emergence of unique structures, remains an area in need of further systematic study to isolate, quantify, and directly assess the biological activity of each form.

For yeast prions, the correlation between in vitro-assembled amyloid and its biological consequences has been firmly established. For example, polymerization of the prion-determining domain (PrD; amino acids 1–254) of the Sup35 protein in vitro can be templated either by in vitro-formed fibrils or cell lysates from a [*PSI*^+^] strain [55], and prion fibrils purified from yeast lysates share similar morphologies with those produced in vitro using ex vivo-derived templates [56]. Most definitively, Sup35 PrD fibrils produced in vitro can convert [*psi*^−^] cells to the [*PSI*^+^] state [31], suggesting that the mechanism underlying the templating of these structures is shared to some extent between in vitro and in vivo-derived material. 

Despite these connections, there is significant evidence of condition-dependent structural heterogeneity in vivo and in vitro, which may provide another framework in which to understand differences between ex vivo and in vitro fibrils. For example, Sup35-PrD fibrils assembled at different temperatures in vitro adopt distinct amyloid cores, as assessed by hydrogen–deuterium exchange [32], and induce distinct [*PSI*^+^] variants when transfected into yeast cells [33]. These variants, known as Sc4 and Sc37, are phenotypically similar to those induced by overexpression of Sup35 protein in a [*psi*^−^] strain (i.e., [*PSI*^+^]^Strong^ and [*PSI*^+^]^Weak^, respectively) [33], but a detailed analysis of their kinetic stabilities and amplification rates reveals subtle differences that impact their sensitivity to inhibition by the dominant-negative Sup35 mutant G58D [57]. Thus, although related, the variants dominating in vitro and in vivo appear both structurally and biologically distinct, and the temperature effects in vitro suggest that selective pressure can be applied to influence the dominant species. As is the case for temperature in vitro, applying pressure to this “cloud” of [*PSI*^+^] variants by expressing Sup35 mutants [58] or altering the levels of Hsp104 [59], a molecular chaperone required for amyloid fragmentation, can select for variants whose stability is dependent on those conditions, presumably restricting conformational diversity. 

A similar concept has been proposed in mammals. The cloud hypothesis for PrP^Sc^ describes an intrinsic structural heterogeneity within individual strain isolates. This structural “mutability” allows for competition between distinct population pools and an adaptive capacity through selection for the most advantageous conformations when the replication environment changes [60,61]. The in vivo replication environment therefore integrally defines the structures capable of persevering in vivo. In this context, the need for PMCA and co-factors to produce recombinantly infectious material can be understood from a structural diversity perspective. If some degree of structural mutability is present, the PMCA method effectively creates opportunity for selection at every round during amplification, and the presence of co-factors help define those selective pressures, allowing in vivo-compatible conformations to arise and amplify. The non-infectious nature of the material produced by endpoint in vitro fibrillization or PMCA without co-factors may therefore be defined by structural incompatibility with the in vivo environment or an insufficient degree of heterogeneity allowing adaption to the in vivo environment. Indeed, the use of the structurally more diverse “universal acceptor” PrP^C^ from bank voles can fully restore infectious activity of previously non-infectious material by PMCA with co-factors within three amplification rounds [62], suggesting that the infectious properties of an aggregate population can be modified by imposing greater structural flexibility. Similarly, the “proteolytic selection” mechanism postulates that a range of fibril structures may be accessed in vivo but are selected for or against based on their proteolytic stability, leading to more stable structures, within the range capable of being fragmented to propagate, dominating the population [51]. Consistent with this idea, ex vivo serum amyloid A fibrils are morphologically distinct and proteolytically more stable than those produced in vitro, and these properties can be transferred to serum amyloid A protein if it is seeded with ex vivo fibrils, indicating that this conformation is accessible to the protein but not the favored form under unseeded conditions [63].

In the context of conformational diversity, post-translational modifications (PTMs) also represent a potential structural constraint present in vivo that is not available to direct amyloid assembly in vitro. A range of PTMs are known to affect amyloid-related disease states [64]. For example, sialylation of PrP is emerging as a substantial driver of prion pathogenesis [65] and as such may represent a selective constraint on conformations that can propagate in vivo, potentially defined by a balance between hindering conversion of monomer into protofibrils and protecting assembled protofibrillar structures from destruction by cellular processes [65]. Although the presence of PTMs can explain structural differences between in vitro and in vivo-derived material at the monomer packing level, PTMs on the protofibril surface are also likely to influence protofibril assembly into fibrils. Despite these predictions, direct investigation into how PTMs define the higher-order amyloid architecture observed in vivo is currently lacking, and a systematic analysis of their impact on both structural and phenotypic outcomes would contribute significantly to our understanding of amyloid biology.

Taken together, this evidence suggests that in vitro-assembled structures share key functional similarities with those arising in vivo but are not necessarily directly representative of the predominant in vivo state. While it is unsurprising that these amyloids are not identical given the spectrum of conformational states accessible to prions and prion-like proteins and the multitude of experimental conditions possible in vitro and in vivo, the ability to move between these contexts expands and informs our understanding of the connections between amyloid structures, their regulation, and their associated phenotypes.

## 3. Prion and Prion-like Proteins Access a Range of Aggregate States with the Potential for Biological Impact

Beyond protofibrils, amyloidogenic proteins can adopt numerous other structures, and ongoing uncertainty and debate over the role of amyloid itself in pathology for many diseases omits them from being classified as *bona fide* amyloidoses, although this may change in the future [66]. Instead, cellular toxicity of clinically significant amyloids such as the Alzheimer’s disease-associated Aβ and the Parkinson’s disease-associated α-syn amyloids has been linked to smaller protofibril and pre-amyloid oligomeric species [67]. These oligomeric species, whether on or off the pathway to fibrilization, have been heavily implicated in cellular toxicity through a common mechanism of membrane disruption seen across multiple amyloid-forming proteins. α-syn fibrils, for example, have been observed to release oligomers both in vitro and in cell culture that can trigger cell dysfunction and permeabilize neuronal membranes [68]. Furthermore, recombinant oligomeric soluble Aβ can adopt various morphologies found to be toxic and sometimes membrane destabilizing [69], including annular [70], sphere-like oligomers termed β-balls [71] and wreath-like structures [72]. The toxicity of these species may be common to their structures as opposed to the specific protein forming them, as protofibrils produced from non-disease-forming proteins also demonstrate cellular toxicity [73], and anti-oligomer antibodies raised against an Aβ analogue are able to neutralize toxicity across multiple amyloid models [74].

Much of the evidence for cytotoxic oligomers is, however, provided by recombinant in vitro-formed fibrils, with examples from ex vivo amyloid isolated from diseased tissue remaining sparse. In the case of Alzheimer’s disease, however, the development of oligomer-specific antibodies has helped provide some evidence that toxic oligomeric states exist in vivo and associate with disease. The soluble fraction of Aβ isolated from human brain, for example, correlates more strongly with disease severity than the insoluble fraction [75], and in some cases direct observations of oligomeric morphology can also be made [76,77]. Soluble Aβ dimers isolated from the brains of human patients may also disrupt synapse structure, function, and long-term potentiation (LTP) in a mouse model [78]. These oligomers were also found to rapidly form larger synaptotoxic protofibril structures [79], illustrating how these observed assembly states are ultimately still only a snapshot of what is most likely a highly dynamic and complex system in vivo.

In the case of PrP, infectivity determined by animal bioassay or cell culture models provides a much more convenient means to assess the biological relevance of lower molecular weight species. These types of experiments have also suggested that smaller fibril species are the most infectious, have the greatest conversion activity, and lead to earlier disease [80,81]. It should, however, be noted that these studies normalize samples by total protein not particle number, and if the dominant driver of infectivity is the number of surfaces capable of monomer conversion, the number of such particles is more biologically relevant than their proportion of protein in this state. Thus, developing methods to assess this key parameter are essential to accurately assess biological impact. Furthermore, PrP^Sc^ fibril infectivity and toxicity can be decoupled from one another. Recently, PrP^Sc^ prion fibrils purified from brain homogenate were determined to be highly infectious by the automated scrapy cell assay (ASCA) but not neurotoxic in primary neuronal cell culture [7]. The detergent treatment associated with their purification was found responsible for this latter effect, as treatment of infectious crude brain homogenate with detergent also destroyed toxicity but not infectivity [7]. Further investigation into the effects of detergent treatment on the biophysical properties of these assemblies may offer a unique opportunity to directly assess how fibril architecture impacts biological activity.

Protofibrils, once formed, can also assume a diversity of higher-order architectural morphologies, where multiple protofibrils composed of the same protein assemble laterally through helical twisting and helical turns to form fibrils of varied thicknesses [34,82,83] (see Figure 1). The clarity of ex vivo structural insight into these types of assemblies is unprecedented and represents a substantial leap in our understanding. Recent progress in cryo-EM techniques and isolation methodologies now allow atomic-resolution determination of structure and morphology of ex vivo assemblies from diseased tissues. Through careful isolation of fibrils from human and animal tissues, it has been suggested that amyloid polymorphisms exist ex vivo as differences in higher-order protofibril assemblies (see Table 1). Fibrils can be composed of many protofilaments and vary with disease [84], as reported for Alzheimer’s-derived Aβ fibrils [53], AA amyloidosis from human kidney [85] or mouse models [51], tau fibrils from Alzheimer’s disease patients [86], multiple system atrophy (MSA) α-syn fibrils [50], and RML PrP^Sc^ fibrils isolated from mouse models expressing PrP with or without the GPI anchor [41,49,87]. However, care should also be taken when interpreting fibril morphologies in the context of width measurement and protofibril number, as some structural models allow for single monomers to span the full measured width of a fibril instead of two monomer stacks contributed by two protofibrils [88,89]. This nuance has been attributed to flexible hinge regions allowing such conformation flexibility for PrP^Sc^ fibrils [88], to various structural breaks within the same fibril stack for amyloid light chain fibrils from A amyloidosis-diseased human heart tissue [90], and to parallel in-register intermolecular β-sheet (PIRIBS) structure of single monomers [89].

The presence of higher-order fibril architecture ex vivo from diseased tissues across multiple disease states has been interpreted as strong evidence that these structures are biologically relevant and present in vivo. Nonetheless, these structures are acquired after extensive isolation procedures which vary considerably among publications and should be taken into consideration when comparing observations (Table 1). The manner through which tissues are handled and fibrils are purified will influence their observed structures. For example, mechanical lysis methods and sonication could potentially shear large assemblies; detergent treatments likely disassociate interactions; protease treatments select for specific species; and sedimentation strength affects aggregate morphology by forcing aggregates together into larger disordered clumps, reducing the individual particle number [41]. It is evident that many of these variables differ between studies despite 2–3 protofibril structures remaining prevalently observed across multiple isolation methods. This consistency lends some weight to the robustness of these observations, but it should also be stressed that the ex vivo isolation procedure may still alter the true in vivo state. For some proteins such as PrP, it is possible to monitor infectivity throughout purification procedures to ensure relevant species are not lost [91]. However, infectivity may not always correlate with toxicity [7], suggesting that key clues to the true in vivo state may still be lost. At best, isolation of these various aggregation states from smaller oligomeric species to larger, higher-order protofibril architectures are ultimately an ex vivo snapshot of a complex and dynamic system. Asking how these observed assembly states contribute to the biology occurring in vivo is a key question moving forward, and the answer will require the development of new methods to probe structure in vivo. 

In the case of the [*PSI*^+^] prion, Sup35 exists as a heterogeneous mixture of aggregate sizes in vivo as determined by fractionation of [*PSI*^+^] lysates on sucrose gradients and by migration by semi-denaturing detergent agarose gel electrophoresis (SDD-AGE) [5,9]. The detergent resistance of the Sup35 aggregates in [*PSI*^+^] lysates, in contrast to detergent-sensitivity of aggregates formed by Sup35 overexpression in [*psi*^−^] strains in the absence of prion conversion [58], mirrors that of Sup35 PrD protofibrils assembled in vitro [92], further suggesting prion-associated aggregates are themselves amyloid. Analysis of the sucrose gradient fractions by SDD-AGE reveals that each progressively dense fraction of native complexes contains progressively larger detergent-resistant aggregates, revealing a direct correlation [9]. However, the size of the Sup35 aggregates analyzed by sucrose gradient was estimated to be ~35-fold larger than those analyzed by SDD-AGE [9], indicating these species are related but distinct. Indeed, high molecular weight Sup35 aggregates sediment to less dense fractions on sucrose gradients in the presence of SDS [9,93], and when treated with the same detergent, the ability of purified Sup35 aggregates to induce [*PSI*^+^] when transfected into a [*psi*^−^] strain also increases [5], consistent with the higher transmissibility of smaller complexes associated with the prion state [81,94].

Two non-mutually exclusive models have been proposed to explain these differences. First, Sup35 protofibrils are associated with partner proteins and other macromolecular complexes, such as ribosomes, in vivo, through SDS-sensitive interactions. Consistent with this idea, immunocapture of Sup35 from [*PSI*^+^] lysates and analysis by SDS-PAGE or RT-PCR reveals co-captured proteins and ribosomal RNA [95]. Second, SDS-resistant Sup35 protofibrils are assembled into higher-order, SDS-sensitive native complexes by lateral association [9]. Evidence for such a higher-order Sup35 complex in vivo, as well as for other prion-forming proteins including the HET-s prion of *Podospora anserina* [6], has been revealed by electron microscopy (EM) methods. For example, morphological size measurements of Sup35 aggregates immunocaptured from [*PSI*^+^] yeast lysates are consistent with the bundling of fibrils even in the presence of SDS [5], and morphologically distinct filaments and bundles of them have been immunoprecipitated from [*PSI*^+^] yeast lysates in the presence of Tween-20 [96]. Such architecture has also been observed in vivo where Sup35 aggregates are visualized as dot-like by fluorescence microscopy in yeast strains expressing a Sup35 fusion to the green fluorescent protein (GFP) and as bundles of laterally associated fibrils by rapid-freeze EM [8], where interfibrillar structure has been attributed to possible chaperone binding [12]. Others have, however, observed that ex vivo [*PSI*^+^] aggregates purified in the absence of detergent are present as short, single filaments and ring-like oligomers [56], although lateral association was observed in a prion variant-dependent manner for fibrils assembled in vitro from purified PrD using ex vivo-derived [*PSI*^+^] aggregates as a template [34]. 

While there is clear evidence from EM analysis of yeast cells that these kinds of higher-order architectures can be present in vivo, they have only been observed with the expression of the isolated PrD to levels well beyond that of the endogenous protein [6,8,12,14,97]. Thus, it is unclear if these structures are the product of biologically significant regulated events or an artifact of driving cells too far outside of their proteostatic capacity. Similarly, they have only been observed in lysates treated with detergent and/or concentrated by centrifugal force [5,34,56,96], which may promote association by stripping away partner proteins that shield interaction surfaces or increasing the concentration of fibrils, respectively. To add further evidence to these early observations, there is a clear need for supplementary methods that allow the biophysical characterization of in vivo [*PSI*^+^] aggregates under endogenous expression conditions, the direct comparison of the species observed under native and detergent conditions, and the correlation of these observations to biological outcomes. With many experimental conditions to induce transitions between the [*psi*^−^] and [*PSI*^+^] states having been identified, such approaches have the potential to directly correlate higher-order structure to biological impact.

## 4. De Novo Induction of [*PSI*^+^] Is Associated with Higher-Order Sup35 Architecture of Unknown Origin and Function

The visualization of many prion proteins using fusions to a fluorescent reporter have provided real-time analysis of structural dynamics in vivo and may begin to address the biological impact of distinct amyloid assemblies [29,98,99,100]. These studies reveal that prion proteins adopt a diffuse distribution in cells lacking the prion state and coalesce into cytoplasmic foci in its presence. The transmission of the prion state to non-prion Sup35 protein has been directly visualized by mating a [*psi*^−^] cell expressing Sup35-GFP and a fluorescent reporter of the [*PSI*^+^] phenotype to a [*PSI*^+^] cell expressing untagged Sup35. Upon cell fusion, the prion phenotype emerges concomitantly with Sup35 incorporation into foci at the single-cell level [101,102]. These foci bind to the amyloid-selective dyes thioflavinS and Congo red both in vivo [103] and ex vivo [96], providing direct evidence of their amyloid character. 

Fluorescence microscopy studies have been utilized extensively as a tool to monitor changes in the native aggregation state of Sup35 in vivo during [*PSI*^+^] induction. In these analyses, Sup35-GFP or PrD-GFP fusions are overexpressed at high levels, and the emergent aggregation patterns observed by fluorescence microscopy in live cells. Induction of [*PSI*^+^] by this method is associated with the presence of Sup35 aggregates detected as large foci and/or ring-like structures, which have been interpreted as filament-containing complexes [104,105,106], with similar structures also observed for the HET-s (PrD)-GFP prion protein [107]. Evolution in the type of patterning has been observed in time courses, including transitions among line, ring, and mesh-patterned aggregation [108] and extension of dots to the cell periphery to form rings [109]. Moreover, biased transition between certain pattern types has also been proposed [110], suggesting that these observable aggregation patterns represent intermediate states in an aggregation pathway. 

These changes in fluorescence patterning may result from distinct events that are not necessarily mutually exclusive. First, they appear to represent transitions in high-order architectures, as dots, rods, and ring structures are all found to be composed of protofibrils 20–30 nm in diameter assembled laterally into higher-order architectures by rapid-freeze EM [8,14]. These studies suggest that a core protofibril component is common to all of these structures, and that the length and extent of association of these protofibrils determines the morphology of these large fluorescent patterns. 

Second, transitions among these structures may reflect changes in sub-cellular localization, exacerbated by alterations in cellular environments and the state of stress over the experimental regime. For example, proteasome inhibition, elevated temperature, and other stresses induce misfolded proteins, including the amyloidogenic proteins associated with the [*PSI*^+^], [*PIN*^+^] and HET-s prions [108], to form single foci at the insoluble protein deposit (IPOD) or juxta nuclear quality control compartment (JUNQ) [111], while glucose limitation promotes packaging of both monomeric and aggregated Sup35 in extracellular [112] as well as periplasmic vesicles [113]. Intriguingly, the Sup35 PrD interacts with cytoskeletal components and is at the yeast aggresome, [114] and the treatment of a [*PSI*^+^] strain with LatA makes PrD-GFP adopt a diffuse pattern [115]. Together, these observations may suggest that the ring-like patterns observed at the cell periphery could be associated with trafficking to the periplasmic space. 

Third, changing environmental conditions can also induce proteins to undergo liquid–liquid phase separation (LLPS), where they partition into dense-phase, liquid-droplet-like structures at greater local concentrations and yet are still capable of free exchange with the cytoplasm [116,117]. Although multiple mechanisms of LLPS formation have been proposed, concentration and intrinsically disordered structural regions of contributing proteins have emerged as strong drivers of these processes. Not surprisingly, LLPS has been observed for recombinant protein and in cell-based models for the clinically relevant proteins FUS, TDP-43, α-syn, Tau, and PrP [116]. For Sup35, the PrD exhibits structurally disordered behavior in vitro before fibrilization by single molecule FRET studies [118] and assembles into oligomeric species with molten characteristics by fluorescence anisotropy [119]. Correspondingly, energy limitation and pH change induces Sup35-GFP to reversibly form SDS-sensitive foci in vivo and in vitro [120]. Understanding the biological impact of LLPS is an emerging field [117,121,122], and evidence is only beginning to be accumulated for the role of LLPS in yeast prion biology. Nonetheless, the possibility that these structures can be accessed in vivo must be considered when interpreting the formation of foci in vivo, especially under conditions of high expression and where the biophysical nature of these assemblies is not investigated. 

At present, differentiation among these scenarios—higher-order assembly into structurally independent prion architectures, trafficking to and localization at sub-cellular compartments, and LLPS—is limited by a lack of methodology to biophysically interrogate these in vivo structures and by nonuniformity in experimental conditions, such as expression level, duration, and metabolic state, which are all known to have outsized effects on prion dynamics that are challenging to resolve. Despite these complications, correlative studies suggest conditions giving rise to these higher-order architectures coincide with those necessary for [*PSI*^+^] induction [14,109]. For example, when mothers and their daughter progeny are carefully tracked, dots and rings only appear to be present when [*PSI*^+^] is induced, and gene mutations that prevent [*PSI*^+^] induction also do not support the formation of ring structures [123]. Ring structures have been observed to extend from mother to daughter cells through the bud neck [106,110], but appear unable to transmit [*PSI*^+^] without continued high levels of Sup35-PrD [14]; instead, smaller, more mobile dot-like foci occur only in the progeny of mother cells with these structures [14,106,110], 

While the appearance and evolution of these structures clearly correlate with the emergence of the [*PSI*^+^] prion, the complexity of these dynamics cannot be currently reduced to a molecular mechanism. Indeed, conditions that lead to [*PSI*^+^] appearance, such as Sup35 overexpression and Hsp104 inhibition, correlate with transition into laterally associated multi-fibrillar structures as observed by cryo-EM [8,14]. It is, however, unclear if these states are true, stable protofibrillar architectures, how they relate to the observed transitions between various fluorescent patterns and what direct role, if any, they play in prion appearance. The selective pressures that favor the emergence of distinct prion variants mentioned above and mutations known to impact the efficiency of [*PSI*^+^] appearance coupled with parallel analyses of structural dynamics may offer experimental entryways to gain this insight.

## 5. Assembly State Impacts Stability of the Prion State

At a minimum, a stable prion state requires the formation of an aggregate capable of templating its continual growth by interaction with and conversion of native-state protein, being fragmented to create multiple complexes, and being transmitted to other cells (Figure 2). The minimum transmissible unit is defined genetically by the transmission of the prion phenotype and is known as a propagon [124,125]. What organizational state do [*PSI*^+^] particles adopt in vivo during this time of stable equilibrium, and how does it change during the process of prion loss? 

The presence of visible foci alone may phenotypically indicate the prion state, but these structures do not necessarily represent the propagon. Fluorescence microscopy methods and their limit of detection have improved over time, so the interpretations derived from the presence or absence of visible foci should be taken in context of the probable limits of detection at the time of publication. In addition, previous studies have visualized different fusion proteins expressed to different levels and in different metabolic contexts, complicating interpretations. Nonetheless, multiple studies combining genetic and microscopic analyses indicate that changes in the sizes of visible Sup35 foci impacts [*PSI*^+^] inheritance. 

First, fluorescent species with higher rates of mobility are found in daughter cells by fluorescence correlation spectroscopy (FCS) and fluorescence loss in photobleaching (FLIP) experiments [94,126], and aggregates isolated from daughter cells migrate faster by SDD-AGE, suggesting the smallest aggregates are preferentially transmitted [94]. Second, more intense Sup35 foci emerge when [*PSI*^+^] becomes unstable upon Sup35 overexpression [94,108] or in the presence of chaperone mutants or inhibition [13,127,128]. In the latter case, inhibition of the molecular chaperone Hsp104 with guanidine HCl (GdnHCl) in yeast strains overexpressing Sup35 PrD-GFP leads to the emergence of dot [14], rod [129], and ring [106] structures similar to those identified during [*PSI*^+^] induction, suggesting that these structures arise when chaperone activity becomes limited. As was true during induction, the rod-shaped foci formed during GdnHCl treatment are composed of longer filaments laterally associated in parallel to each other [8]. By comparison, SDS-resistant Sup35 aggregates shift to more slowly migrating species by SDD-AGE, native Sup35 aggregates shift to more dense fractions on sucrose gradients [9,13,94,130,131], and the mobility of fluorescent foci in vivo is reduced as assessed by FCS or FLIP [94,101,129]. While these observations reveal a direct correlation among fibril length, native complex size, and transmissibility, the initial form of the native complex (i.e., protofibrils or bundles of protofibrils and partner-protein associations), whether this structure changes upon chaperone inhibition, and if such a change is necessary for prion instability are currently unknown.

In contrast to these correlations, overexpression of Hsp104, which induces [*PSI*^+^] loss [127], decouples the behavior of Sup35 fluorescent foci and SDS-resistant aggregates. While the former disappear [11], the latter have reduced migration by SDD-AGE [9]. Three mechanisms have been proposed to explain [*PSI*^+^] loss during Hsp104 overexpression: fragmentation inhibition [132], transmission inhibition [133], and dissolution [11]. However, current models struggle to explain the differential behavior of Sup35 fibrils and native complexes using available assays (see Table 2); although it is important to note that some experiments have been performed under conditions where Sup35-GFP foci are only visible when cells are transferred to water [11,134,135] or aged in culture [13], questioning the physiological relevance of these observations. Thus, repeating these analyses in parallel using a fusion protein and conditions more reflective of the endogenous protein state is absolutely necessary to draw robust conclusions.

Despite the limitations on currently available observations, transition between states of higher-order architecture during Hsp104 overexpression could help explain these data. For example, increasing Hsp104 levels has been shown to titrate Ssa1/2 from fluorescent foci formed during Sup35 PrD overexpression [132], and Ssa1/2 has been implicated in the alteration of [*PSI*^+^] stability under certain conditions [136,137]. The Sup35:Ssa1/2 interaction is SDS-sensitive and, therefore, presumably at the fibril surface. It is conceivable that the Sup35:Ssa1/2 interaction could regulate fibril:fibril association and thereby higher-order architectures (see Figure 3). Although the effect of SDS treatment or Hsp104 overexpression on the fractionation of Ssa1/2 from [*PSI*^+^] lysates on sucrose gradients has not been assessed, structural spacing between fibrils in vivo as determined by cryoEM does indeed accommodate the possibility of these kinds of interactions [12], and several observations suggest that Ssa1/2 impacts Sup35 aggregate architecture. First, overexpression of Ssa1/2 increases the size of SDS-resistant species by SDD-AGE and the fluorescence intensity of foci formed by a Sup35-GFP fusion that can functionally replace Sup35, but the size of those aggregates as measured by sucrose gradient fractionation decreases [10]. Second, limitation of Ssa1/2 disrupts laminar arrangement of Sup35-PrD into bundles [12], and Sup35-PrD foci are larger in an *SSA1*-*21* mutant background under native conditions, although this effect appears to be independent of the size of detergent-resistant species observed by SDD-AGE which remain unchanged [13]. The dynamics of detergent-resistant and detergent-sensitive complexes may therefore be decoupled in the presence of the *SSA1*-*21* mutation. While the effects are complex and complicated by variations in experimental conditions, including fusion proteins, expression levels, and metabolic states, these observations are intriguing and could represent a true measurable and regulated transition between higher-order architectural states with biological implications. Indeed, if titration of Ssa1/2 induces Sup35 complexes to adopt a larger, less mobile form, these species would rapidly lose transmissibility, potentially explaining the faster kinetics of curing by Hsp104 overexpression in comparison with Hsp104 inhibition which requires continued fibril growth to exceed the size threshold for transmission [94,101]. Complementary and parallel analyses allowing direct comparisons of the behavior of the same Sup35 protein at different scales upon overexpression of Hsp104 or Ssa1/2 could provide an avenue to reveal this insight.

## 6. Conclusions

Evidence is compelling that prion complexes can adopt high-order assemblies in vivo and that these assemblies appear structurally ordered. The biophysical nature of these higher-order architectures or their role in amyloid biology is, however, still unclear. To resolve this unknown, a concerted effort to isolate, quantify, and assess the biological impact of distinct protein assemblies in parallel is crucial. As noted throughout this review, an ensemble of approaches and methodologies have been developed, but these are most frequently used in isolation of one another, raising questions as to how the methods relate to one another and the true nature of the aggregation states that they have described.

Integrating current studies to reveal the correlation between amyloid assemblies and their biologic impact may, however, provide pathways to resolving the existing ambiguity. Specifically, unique conditions are known to promote the assembly of distinct structures in vitro. The biologic activity of these structures should be assessed in parallel using existing cell-based and/or animal models, provided that the abundance of each can be determined by quantifying particle number. Similarly, in vivo models can be subjected to conditions and/or treatments that change assembly state. By using common protocols to extract these species in parallel, direct comparison of their structures using existing methods becomes possible. Carefully controlled application of existing methodologies in parallel and under the same experimental treatment regimens will undoubtably help relate these observations and define the true nature and biological significance of these unique higher-order assemblies.

## Figures and Tables

**Figure 1 viruses-14-01635-f001:**
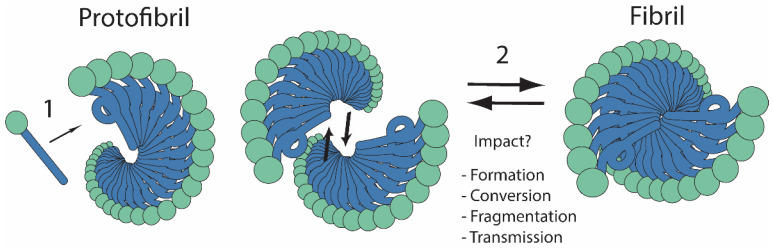
Amyloid higher-order architecture: an assembly state of unknown significance. Protofibrils are assembled from monomeric amyloidogenic proteins through direct association (arrow 1) of the same domain of the protein (blue) among adjacent monomers. This association results in the conformational conversion of the domain (linear to corkscrew) to form a b-sheet rich amyloid core with the non-amyloid domains of the protein (green circles) arrayed on the surface of the protofibril. These protofibrils can then laterally associate into higher-order fibril architectures (arrow 2). The prevalence of these assembly states or how they influence amyloid biology is poorly understood.

**Figure 2 viruses-14-01635-f002:**
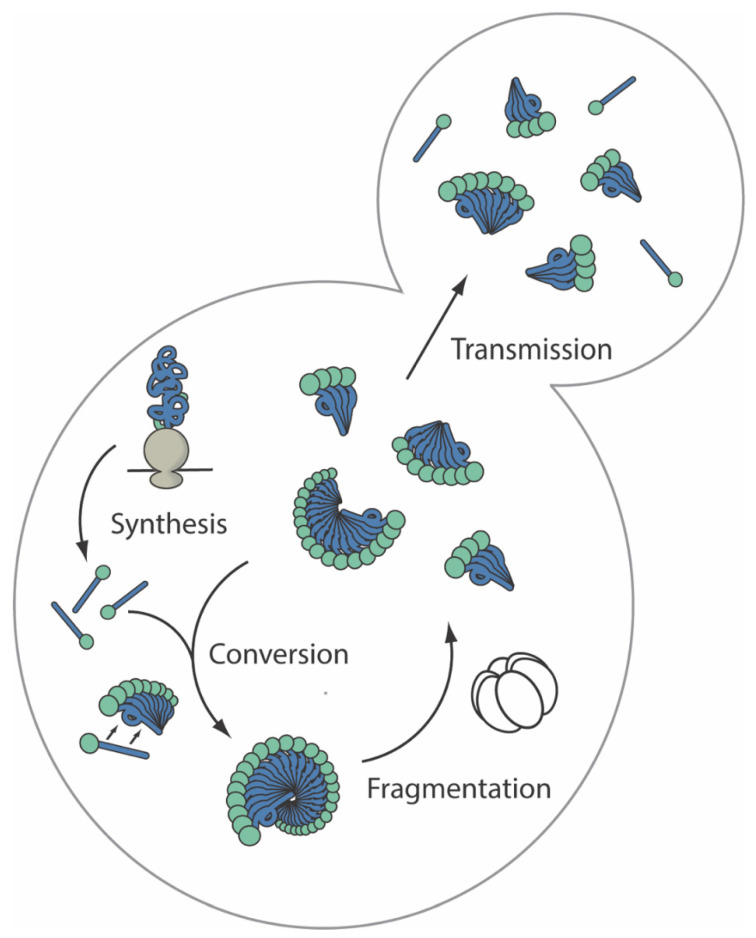
The [*PSI*^+^] prion propagation cycle in the budding yeast *Saccharomyces cerevisiae*. Newly synthesized Sup35 (green and blue ball and stick) from translating ribosomes (gray) forms prions via interaction of its prion-determining domain (blue), leading to its conformation conversion (linear to corkscrew) and the display of its translational release domain (green circle) on the surface of the protofibril. Conversion occurs de novo at low frequency when overexpressed (not shown) or through association with existing linear prion complexes (ball and corkscrew wheels) via the prion-determining domain (blue) through templated conversion at particle ends (double arrow), increasing particle size and reducing particle mobility. Fragmentation of particles by the disaggregase Hsp104 (white hexamer of ellipses) in cooperation with the Hsp70/40 (Ssa1/2) chaperone system (not shown) increases particle number, amplifying the abundance of conversion surfaces available for templating, decreasing the size of these particles and increasing their mobility (green and blue ball and corkscrew wheel fragments). Small mobile particles transmit into daughter cells at cell division maintaining the prion in the cell population.

**Figure 3 viruses-14-01635-f003:**
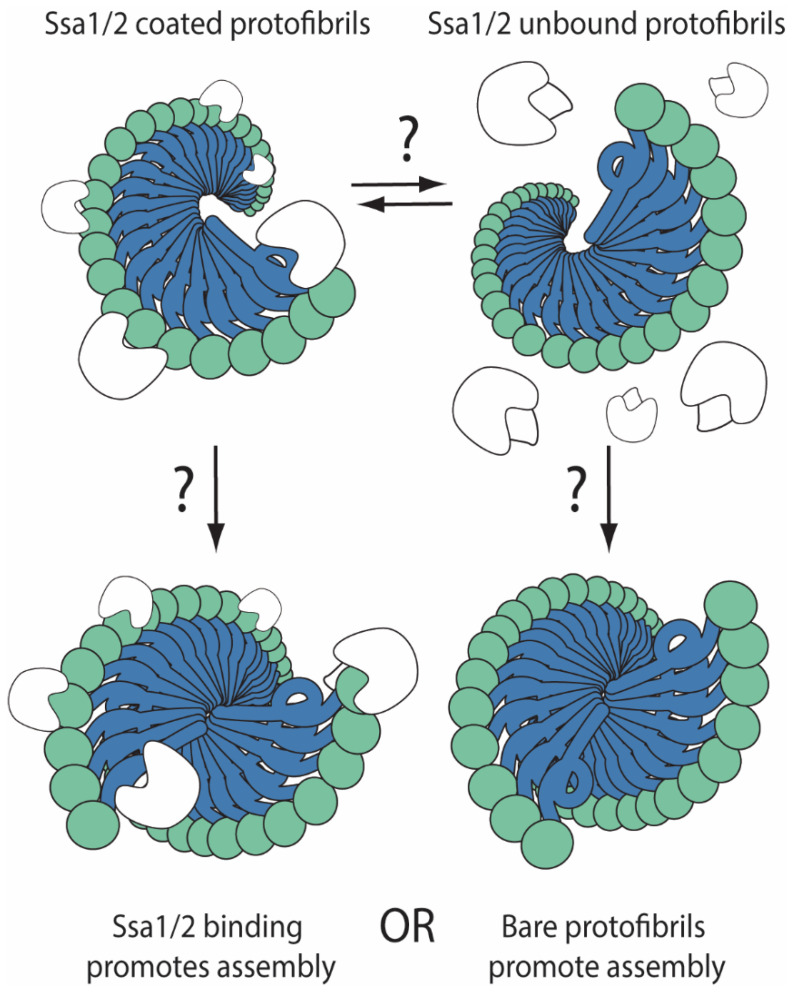
Is [*PSI*^+^] higher-order architecture controlled by chaperone binding? Hsp70 (Ssa1/2; white) appears to be associated with [*PSI*^+^] prion complexes (green and blue ball and corkscrew wheel stacks); however, what determines association and disassociation from these complexes or how this impacts protofibril association into higher-order fibril architectures is unclear.

**Table 1 viruses-14-01635-t001:** Comparison of ex vivo fibril isolation methods from diseased tissues and the resulting fibrillar morphologies determined by cryo-electron microscopy.

Amyloid Source	Disruption Method	Extraction Type	Enzymatic Digestion	Detergent	Max. Sediment	Observed Fibril Morphology By EM	Reference
**Prion Disease–PrP^Sc^**							
RML PrP^Sc^ fibrils	Homogenized (tissue grinder)	Selective protein precipitation: NaPTA	Pronase Benzonase	Sarkosyl	16,100× *g*	2 protofibrils per fibril	[49]
GPI-anchorless PrP (27-30) RML PrP^Sc^ fibrils from mouse brain	Dounce homogenization High salt	Detergent resistant membrane (DRM): Brij-96 and high salt: NaCl extraction	Benzonase PK	Brij-96 Sarkosyl	Approx. 200,000× *g*	2 protofibrils per fibril	[87]
aRML fibrils from mouse brain or 263K PrP^Sc^ from hamster brain	Dounce homogenization Sonicated	Water	PK	Sulfobetaine	225,000× *g*	1 protofibril per fibil PRIBS-based model for PrP^Sc^ in which a single PrP molecule spans the entire fibril width to give an asymmetric cross-section	[89]
**AL Amyloidosis–Amyloid Light-Chain (AL)**							
AL amyloidosis human heart tissue	Kontes pellet pestle	Water	Collagenase	no	3100× *g*	i and ii polymorphism types by fibril width, pitch and width of crossover	[52]
Systemic AL amyloidosis human heart tissue	Kontes pellet pestle	Water	Collagenase	no	3100× *g*	1 protofibril per fibril	[90]
**AA Amyloidosis–Serum Amyloid A (SAA)**							
AA amyloid fibrils from human kidney	Kontes pellet pestle	Water	Collagenase	no	3100× *g*	2 protofibrils per fibril	[85]
AA amyloid fibrils from mouse liver	Homogenized with scalpel	Water	Collagenase	no	3100× *g*	2–3 protofibrils per fibril	[51,63]
**Multiple System Atrophy (MSA)–** **α-syn**							
α-syn fibrils from MSA human brain	Homogenized	Sarkosyl-insolubility	no	Sarkosyl	166,000× *g*	2 protofibrils per fibril	[50]
**Alzheimer’s Disease (AD)–** **Aβ and tau**							
Aβ fibrils from human AD brain	Homogenized (scalpel)	Water	Collagenase	no	12,000× *g*	1–3 protofibrils per fibril	[53]
Tau fibrils from human AD brain	Homogenized (polytron)	Sarkosyl-insolubility Superose 6 increase column Vivacon 500 conc.	Pronase	Sarkosyl	100,000× *g*	2 protofibrils per fibril Paired helical filaments (PHFs) and straight filaments (SFs) each composed of common protofibril structure	[86]

**Table 2 viruses-14-01635-t002:** Comparison of Hsp104 overexpression studies and observations. Use of ? denotes where the indicated properties were not measured. ✓ denotes property was observed. X denotes property was not observed.

Hsp104	Sup35	Curing	Time Point Post Induction at Which Observations Made	*In Vivo* Observation	*Ex Vivo* Observations (Native)	*Ex Vivo* Observations (Detergent Present)	Reference
Multi copy plasmid	WT	?	Stable expression	?	Size shift to smaller fractions when sedimented through 30% sucrose	?	[131]
GAL inducible plasmid for 48 h	NM-Gfp driven from CUP inducible plasmid for 48 h	✓	48 h	Ring aggregate patterns	?	?	[106]
Multi copy plasmid	WT	?	Stable expression	?	?	Size shift larger we fractionated by SDD-AGE. Increased monomer signal	[9]
CUP inducible plasmid for 8 h	NM-Yfp driven from GAL inducible plasmid for 10 h	?	10 h	Sup35NM-Yfp les mobile as determined by FLIP and localized to the cytoplasm	?	?	[132]
GAL inducible plasmid	WT	✓	4 generations	Propagons malpartition to mothers	Soluble pool increase	No size shift when fractionated by SD-AGE	[133]
GAL inducible plasmid TET inducible plasmid	Sup35N-Gfp-MC	GAL ✓ TET X	GAL 1 generation TET 12 generations	Loss of detectable foci that are recovered by 1 h water treat	?	?	[135]

## Data Availability

Not applicable.

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
