# Peer review of "Beyond Amyloid Fibers: Accumulation, Biological Relevance, and Regulation of Higher-Order Prion Architectures"

_viruses, 2022, doi:10.3390/v14081635_

Round 1

Reviewer 1 Report

The review by Naeimi and Serio provides a comprehensive analysis of both the yeast and mammalian prion literature, specifically focusing on how ex vivo, in vitro and in vivo evidence that contributes to current models. The review also makes strong points where research is contradictory, or where more work is needed. This is the first review that I have read to date that has successfully and adequately places both the yeast work and mammalian work into a larger context.  The authors have done a tremendous amount of work with the integration, and this compelling review will be a benefit to the field. 

There is only one major issue with the review (and some minor comments).

Major issues.

Figure 2 – It is difficult to understand what the point is of figure 2.  The text vaguely refers to figure 2, suggesting that  transitions between aggregate structures could define parameters key to prion biology. However, when referring to figure 2, the reader has to come to their own conclusions on how the figure fits within the conclusions of the text. It is recommended to significantly revise this figure (with a robust figure legend that reminds the reader that the hexamers are Hsp104, and what the proposed role Hsp104 plays in this conversion).  Otherwise, the authors should eliminate this figure entirely.

Minor comments.

Line 433.  “In fact, localization of monomeric Sup35 at the IPOD is a pre-requisite to aggregation and induction of [PSI+] (Tyedmers et al., 2010; Arlsan et al., 2015).”  Simple co-localization experiments during Sup35 induction does not mean that localization to the IPOD is a pre-requisite. Other studies suggest that formation happens at the cell periphery (Mathur et al., 2009) or cortical actin patches (Ganusova et al., 2006; Chernova et al., 2011).  Timelapse experiments of newly formed prion aggregates show that Sup35 foci are initially highly mobile and then slow down (Sharma et al., 2017; Lyke and Manogaran, 2017), indicating that the experiments from Tyedmers and Arslan may have been taken at timepoint much down stream from initial formation. Please revise this statement  or integrate the work mentioned above.

The de novo section does not include any of the current Sup35 formation models: cross-seeding and titration.  While it is understandable that this topic can be somewhat controversial, the addition of yeast evidence for heterologous cross seeding (such as Derkatch et al., 2001; Keefer et al., 2017) and titration models (which the author provides compelling evidence; Villai et al., 2020) should be included.  The heterologous seeding model shown by the Baskakov lab (Katorcha et al., 2017) could be used as a mammalian parallel. However, it is unclear if the mammalian field has advanced far enough to provide evidence for titration.   While it is not required to include this section, it would greatly benefit the field if this information was included.

Author Response

Reviewer 1:

  1. Figure 2 – It is difficult to understand what the point is of figure 2. The text vaguely refers to figure 2, suggesting that transitions between aggregate structures could define parameters key to prion biology. However, when referring to figure 2, the reader has to come to their own conclusions on how the figure fits within the conclusions of the text. It is recommended to significantly revise this figure (with a robust figure legend that reminds the reader that the hexamers are Hsp104, and what the proposed role Hsp104 plays in this conversion). Otherwise, the authors should eliminate this figure entirely.

We have revised both figures as noted above and changed the order (i.e. original figure 1 is now figure 2 and vice versa). The new figure 1 now provides a general overview of the review, and figure 2 is specific to Sup35.

  1. Line 433. “In fact, localization of monomeric Sup35 at the IPOD is a pre-requisite to aggregation and induction of [PSI+] (Tyedmers et al., 2010; Arlsan et al., 2015).” Simple co-localization experiments during Sup35 induction does not mean that localization to the IPOD is a pre-requisite. Other studies suggest that formation happens at the cell periphery (Mathur et al., 2009) or cortical actin patches (Ganusova et al., 2006; Chernova et al., 2011). Timelapse experiments of newly formed prion aggregates show that Sup35 foci are initially highly mobile and then slow down (Sharma et al., 2017; Lyke and Manogaran, 2017), indicating that the experiments from Tyedmers and Arslan may have been taken at timepoint much downstream from initial formation. Please revise this statement or integrate the work mentioned above.

The reviewer’s point is well taken; we overly simplified the observations. The statement referenced has been removed, and the paragraph has been further edited for clarity.

  1. The de novo section does not include any of the current Sup35 formation models: cross-seeding and titration. While it is understandable that this topic can be somewhat controversial, the addition of yeast evidence for heterologous cross seeding (such as Derkatch et al., 2001; Keefer et al., 2017) and titration models (which the author provides compelling evidence; Villai et al., 2020) should be included. The heterologous seeding model shown by the Baskakov lab (Katorcha et al., 2017) could be used as a mammalian parallel. However, it is unclear if the mammalian field has advanced far enough to provide evidence for titration. While it is not required to include this section, it would greatly benefit the field if this information was included.

At the suggestion of the editors, we have not added additional information on cross-seeding and titration to the review. We concur that it is outside of the scope of this manuscript given the absence of links to higher-order amyloid architecture for either model.

Reviewer 2 Report

The manuscript "Beyond Amyloid Fibers: Accumulation, Biological Relevance, and Regulation of Higher-Order Prion Architectures" by Naeimi & Serio reviews the current literature on prion propagation in yeast and compares the findings obtained in yeast with studies from other more complex organisms. It describes how amyloid aggregates are formed and propagated, and the influence of the cellular environment (e.g., molecular chaperones) on these processes. The authors specifically address the structural heterogeneity of prions and amyloids and discuss the biological relevance of the various structural variants observed in vitro and in vivo. In addition, the caveats and limitations of studying amyloids formed in vitro and amyloids extracted from cells and tissues are debated. The article is very comprehensive and informative.

However, the following modifications may improve the quality of this article:

Major points

1.     The authors repeatedly refer to the lack of appropriate studies that link observations with a particular amyloid fiber generated in vitro to actual amyloid aggregates occurring in vivo.  It would therefore be very helpful if they could propose some more concrete solutions to this problem in the conclusions. The conclusion as presented leaves the reader unsatisfied and with the feeling that the author is withholding a solution.

2.     The idea that prions exist in vivo as a cloud of different conformation is intriguing and should be elaborated on, as this is certainly very difficult to study with fibrils generated in vitro. There are interesting observations on how the cellular environment selects certain types.

3.     In connection with the above point, the role of the cellular environment in the formation and propagation of certain amyloid structures should be emphasized.

4.     Section 4 is very lengthy especially for readers who are not yeast prion experts. It is unclear exactly what message the authors are trying to convey with the detailed description of rings, dots, rods,... It would seem more appropriate to make this section shorter and more concise.

Minor points

5.     Page 6 line 205: If I am not mistaken, this should read “...be defined by structural incompatibility with the in vivo environment…”

6.     Throughout the text there sometimes is too much space between individual letters, e.g. on page 6 line 191, page 8 line 299 or page 10 line 390.

Author Response

Reviewer 2:

  1. The authors repeatedly refer to the lack of appropriate studies that link observations with a particular amyloid fiber generated in vitro to actual amyloid aggregates occurring in vivo. It would therefore be very helpful if they could propose some more concrete solutions to this problem in the conclusions. The conclusion as presented leaves the reader unsatisfied and with the feeling that the author is withholding a solution.

We appreciate this comment from the reviewer. We have revised each section and rewritten the conclusion to be more specific about experimental paths that we feel will help to clarify the current state of knowledge.

  1. The idea that prions exist in vivo as a cloud of different conformation is intriguing and should be elaborated on, as this is certainly very difficult to study with fibrils generated in vitro. There are interesting observations on how the cellular environment selects certain types. In connection with the above point, the role of the cellular environment in the formation and propagation of certain amyloid structures should be emphasized.

We have slightly expanded our explanation of prion clouds (lines 85-88) in the context of structure and have attempted to highlight how this conformational flexibility could be exploited to provide additional insight throughout the review.

  1. Section 4 is very lengthy especially for readers who are not yeast prion experts. It is unclear exactly what message the authors are trying to convey with the detailed description of rings, dots, rods,... It would seem more appropriate to make this section shorter and more concise.

We appreciate this feedback and have edited the section with these points in mind.

  1. Page 6 line 205: If I am not mistaken, this should read “...be defined by structural incompatibility with the in vivo environment…”

We apologize for this error and have correct it.

  1. Throughout the text there sometimes is too much space between individual letters, e.g. on page 6 line 191, page 8 line 299 or page 10 line 390.

We believe that this spacing results from the right justification added by the journal after submission and have therefore not altered it.

Reviewer 3 Report

Figure 1-3

- It is difficult to understand what the figures mean. A detailed explanation of each figure is required.

- This is a review of various amyloids, but all figures are related to PSI. It is highly desirable that authors should add a figure comparing the properties of each amyloid.

Table 1

- It's not clear what the reference is pointing to. It is not possible to distinguish between Ref 40 and Ref 47 which are common and differentiated. Please clarify

- What is the number of “observed fibril morphology by EM” columns? The table itself does not make sense. The detailed explanation is needed.

Author Response

Reviewer 3:

  1. Figure 1-3 - It is difficult to understand what the figures mean. A detailed explanation of each figure is required.

As noted above, we have revised the figures and figure legends and reordered the figures.

  1. This is a review of various amyloids, but all figures are related to PSI. It ishighly desirable that authors should add a figure comparing the properties of each amyloid.

At the suggestion of the editors, we have not added a new figure. Instead, we have updated the previous figure 2 to an overview and moved it to the first figure. In addition, we have included substantial information on other amyloids in Table 1.

  1. Table 1- It's not clear what the reference is pointing to. It is not possible to distinguish between Ref 40 and Ref 47 which are common and differentiated. Please clarify.

Our apologies for this error. We have updated the table with the appropriate reference to Terry et al. 2019.

  1. What is the number of “observed fibril morphology by EM” columns? The table itself does not make sense. The detailed explanation is needed.

We apologize for this confusion and have updated the column headings for clarity.

Reviewer 4 Report

The review by Naeimi and Serio has several issues that should be addressed before the paper can be accepted. Most important of these issues is the model, presented in figures 1-3. In all three figures Supr35 protein is depicted by a blue rod and a green ball. However, it is not explained what the green ball and blue rod are. Upon Sup35 protein forming a prion the blue rod is depicted in figure 1 as containing a loop, presumably indicating a different fold. Individual loop containing molecules are shown suggesting that the prion form of Sup35 can propagate as a monomeric protein. Although, monomeric prion proteins can exist (as illustrated by Roberts and Wickner, Genes Dev. 2003 17:2083-2087) there is ample evidence that this is not the case for the Sup35p based prion [PSI+]. Prions formed by Sup35 are infectious amyloids. Thermodynamically it is hard, if not impossible, to have a protein monomer with the fold that is present in the in-register parallel beta sheet amyloids that Sup35 protein forms.

The legend of figure 1 states that “Sup35… joins existing linear prion complexes through templated conversion at particle ends.”. What exactly is being templated is not explained in the figure nor is it explained in the text. Neither is it explained how this templating mechanism works. As depicted in the figure there is a tiny interaction surface between Sup35 monomers (at the very end of the rods), it is not explained (in the figure or in the text) what the basis of this interaction is.

The prion particles that are being fragmented by Hsp104 in figure 1 are depicted as micelle like assemblies. There is a voluminous body of publications describing amyloid as the basis allowing Sup35 protein to form a prion. There is a molecular level structure showing how PrP folds to form parallel in-register beta sheet amyloid in the brains of scrapie infected animals. There is ample data showing that Sup35 protein also adopts a parallel in-register beta sheet amyloid fold in [PSI+] cells. There is no data whatsoever associating micelle like assemblies with prions or amyloidosis. For both PrP and Sup35p mass per unit length indicates that there is one protein molecule per 4.7 angstroms of amyloid filament length. This fact is irreconcilable with the depiction of particles in figures 1-3.  

There is no molecular level explanation how prion variants (yeast) or prion strains (mammals) can exist. There is a substantial body of literature showing how parallel in-register amyloid folds allow strains and variants to faithfully propagate.

The paper uses a superfluous number of words describing systems that remain poorly defined. A disproportionate amount of space is dedicated to experimental results obtained using GFP tagged Sup35 protein. In many cases GFP tagged proteins are indicative but not do not represent the actual [PSI+] prion. Sometimes facts are repeated. For instance, in the paragraph starting at line 557 Ssa1/2 are discussed whereas these proteins got ample exposure in the paragraph starting with line 349.

Other comments.

Line 71                  I think that people having published data about [SWI+], [PIN+]/[RNQ+], or [URE3] will object to having their work described as being inferior to that published about [PSI+]. There is no doubt that [PSI+] is the most studied yeast prion.

Line 75                  It should be pointed out that in its prion form Sup35p is not functional and that there thus has to be a balanced cycle.

Line 79                  For readers not familiar with amyloid this section would require a better explanation.

Line 93                  It is not clear that the proteostasis network “regulates” aggregated states of Sup35 protein. It does of course act upon these states.

Line 139                For readers not familiar with prions PMCA will need an introduction.   

Line 146               Schimdt et al Open Biol 2015, 5: 150165 explored around 20,000 conditions and could not find any that worked reproducible.

Line 162               Those amyloid filaments obtained from an organism have been selected whereas those formed de novo in a tube have not.

Line 207                It should be pointed out that the bank vole is very unusual in that it can replicate many prions originating from other species (“universal acceptor”).

Line 212                However, amyloid filaments have to break to propagate!

Line 233                It is not clear from the text where the spectrum of conformations is based on.

Line 240                What exactly do the authors mean with pre-and post-amyloid structures?

Line 263                There is data suggesting that tau amyloid levels correlate better with disease progression than a-beta levels.

Line 273                Are these small oligomeric species different than those discussed in the previous paragraphs?

Line 380                If the structures observed are considered artificial by the authors why do they spend so much time discussing them?

Line 399                For those not intimate familiar with S. cerevisiae this might need a bit more explanation.

Line 550                The Sup35 N-GFP-MC fusion expressed from the Sup35 promoter shows fluorescence without the cells having to be kept in water.

Line 570                The relevance of the SSA1-21 mutant should be explained.

Author Response

Reviewer 4:

  1. The review by Naeimi and Serio has several issues that should be addressed before the paper can be accepted. Most important of these issues is the model, presented in figures 1-3. In all three figures Supr35 protein is depicted by a blue rod and a green ball. However, it is not explained what the green ball and blue rod are. Upon Sup35 protein forming a prion the blue rod is depicted in figure 1 as containing a loop, presumably indicating a different fold. Individual loop containing molecules are shown suggesting that the prion form of Sup35 can propagate as a monomeric protein. Although, monomeric prion proteins can exist (as illustrated by Roberts and Wickner, Genes Dev. 2003 17:2083-2087) there is ample evidence that this is not the case for the Sup35p based prion [PSI+]. Prions formed by Sup35 are infectious amyloids. Thermodynamically it is hard, if not impossible, to have a protein monomer with the fold that is present in the in-register parallel beta sheet amyloids that Sup35 protein forms.

As noted above, we have revised the figures and figure legends and reordered the figures.

  1. The legend of figure 1 states that “Sup35… joins existing linear prion complexes through templated conversion at particle ends.”. What exactly is being templated is not explained in the figure nor is it explained in the text. Neither is it explained how this templating mechanism works. As depicted in the figure there is a tiny interaction surface between Sup35 monomers (at the very end of the rods), it is not explained (in the figure or in the text) what the basis of this interaction is.

As noted above, we have revised the figures and figure legends and reordered the figures.

  1. The prion particles that are being fragmented by Hsp104 in figure 1 are depicted as micelle like assemblies. There is a voluminous body of publications describing amyloid as the basis allowing Sup35 protein to form a prion. There is a molecular level structure showing how PrP folds to form parallel in-register beta sheet amyloid in the brains of scrapie infected animals. There is ample data showing that Sup35 protein also adopts a parallel in-register beta sheet amyloid fold in [PSI+] cells. There is no data whatsoever associating micelle like assemblies with prions or amyloidosis. For both PrP and Sup35p mass per unit length indicates that there is one protein molecule per 4.7 angstroms of amyloid filament length. This fact is irreconcilable with the depiction of particles in figures 1-3.

As noted above, we have revised the figures and figure legends and reordered the figures.

  1. There is no molecular level explanation how prion variants (yeast) or prion strains (mammals) can exist. There is a substantial body of literature showing how parallel in-register amyloid folds allow strains and variants to faithfully propagate.

We have added additional information on the structural basis of prion variants (lines 50-55).

  1. The paper uses a superfluous number of words describing systems that remain poorly defined. A disproportionate amount of space is dedicated to experimental results obtained using GFP tagged Sup35 protein. In many cases GFP tagged proteins are indicative but not do not represent the actual [PSI+] prion. Sometimes facts are repeated. For instance, in the paragraph starting at line 557 Ssa1/2 are discussed whereas these proteins got ample exposure in the paragraph starting with line 349.

We thank the reviewer for this feedback and have edited this section for clarity and conciseness.

  1. Line 71 I think that people having published data about [SWI+], [PIN+]/[RNQ+], or [URE3] will object to having their work described as being inferior to that published about [PSI+]. There is no doubt that [PSI+] is the most studied yeast prion.

We apologize for this misunderstanding and certainly did not intend to imply that studies on the [PSI+] prion are superior to those on other prions. We have substituted “most widely” for “best”.

  1. Line 75 It should be pointed out that in its prion form Sup35p is not functional and that there thus has to be a balanced cycle.

Our previous studies have demonstrated that aggregated Sup35 retains some activity (Pezza et al. 2014). We do agree, however, that replication rates must be balanced to keep the prion state within the level of activity required for function and viability and have made a note of it (line 84-85).

  1. Line 79 For readers not familiar with amyloid this section would require a better explanation.

We have edited this section for clarity.

  1. Line 93 It is not clear that the proteostasis network “regulates” aggregated states of Sup35 protein. It does of course act upon these states.

We agree with this nuance and have changed “regulate” to “modulate”.

  1. Line 139 For readers not familiar with prions PMCA will need an introduction.

We have provided an explanation of PMCA (lines 274-277.

  1. Line 146 Schimdt et al Open Biol 2015, 5: 150165 explored around 20,000 conditions and could not find any that worked reproducible.

We have added a sentence to refer to this work on lines 281-282 and have clarified the text to indicate that generating infectious material from recombinant material requires cofactors and PMCA.

  1. Line 162 Those amyloid filaments obtained from an organism have been selected whereas those formed de novo in a tube have not.

We have added a sentence (line 305-306) to clarify the importance of selective pressure in the emergence of distinct structures.

  1. Line 207 It should be pointed out that the bank vole is very unusual in that it can replicate many prions originating from other species (“universal acceptor”).

We have included this information (line 359)

  1. Line 212 However, amyloid filaments have to break to propagate!

This point has been clarified on line 365.

  1. Line 233 It is not clear from the text where the spectrum of conformations is based on.

This point has been clarified through expansion of the discussion on prion clouds.

  1. Line 240 What exactly do the authors mean with pre-and post-amyloid structures?

We apologize for this confusion and have removed that terminology.

  1. Line 263 There is data suggesting that tau amyloid levels correlate better with disease progression than a-beta levels.

While we agree with the reviewer, we were making a separate point about the link between structure and biological outcome for individual proteins. As such. We have not changed the text in response to this point.

  1. Line 273 Are these small oligomeric species different than those discussed in the previous paragraphs?

We have removed this terminology and clarified the text.

  1. Line 380 If the structures observed are considered artificial by the authors why do they spend so much time discussing them?

While we appreciate this point by the reviewer, our goal was to synthesize the current state of knowledge in the field on amyloid higher-order structures. We believe that current knowledge suffers from a lack of integration of distinct approaches. While some conditions may be artificial, they can be useful in clarifying the link between structure and biological outcome, as we have tried to highlight throughout the text.

  1. Line 399 For those not intimate familiar with S. cerevisiae this might need a bit more explanation.

Additional text has been added to clarify these experiments.

  1. Line 550 The Sup35 N-GFP-MC fusion expressed from the Sup35 promoter shows fluorescence without the cells having to be kept in water.

Under conditions of curing, these extreme conditions are required to see foci. We have clarified this point in the text.

  1. Line 570 The relevance of the SSA1-21 mutant should be explained.

We have added text to clarify as noted above.

Round 2

Reviewer 3 Report

All comments made by the authors, this study publish with last form.

Reviewer 4 Report

The revision of the review by Naeimi and Serio is a greatly improved version of the original manuscript. The authors have addressed all the major points that I had pointed out. I thus would recommend publication of the manuscript.